# Exciton-driven change of phonon modes causes strong temperature dependent bandgap shift in nanoclusters

Franziska Münzer[1], Severin Lorenz[1], Jiwoong Yang[2,3], Taufik Adi Nugraha[4], Emilio Scalise [4], Taeghwan Hyeon [2,5✉], Stefan Wippermann [4✉] & Gerd Bacher [1✉]

The fundamental bandgap $E_g$ of a semiconductor—often determined by means of optical spectroscopy—represents its characteristic fingerprint and changes distinctively with temperature. Here, we demonstrate that in magic sized II–VI clusters containing only 26 atoms, a pronounced weakening of the bonds occurs upon optical excitation, which results in a strong exciton-driven shift of the phonon spectrum. As a consequence, a drastic increase of $dE_g/dT$ (up to a factor of 2) with respect to bulk material or nanocrystals of typical size is found. We are able to describe our experimental data with excellent quantitative agreement from first principles deriving the bandgap shift with temperature as the vibrational entropy contribution to the free energy difference between the ground and optically excited states. Our work demonstrates how in small nanoparticles, photons as the probe medium affect the bandgap— a fundamental semiconductor property.

[1] Werkstoffe der Elektrotechnik and CENIDE, Universität Duisburg-Essen, Bismarckstraße 81, 47057 Duisburg, Germany. [2] Center for Nanoparticle Research, Institute for Basic Science (IBS), Seoul 08826, Republic of Korea. [3] Department of Energy Science & Engineering, Daegu Gyeongbuk Institute of Science & Technology (DGIST), Daegu 42988, Republic of Korea. [4] Max-Planck-Institut für Eisenforschung, Max-Planck-Strasse 1, 40237 Düsseldorf, Germany. [5] School of Chemical and Biological Engineering, and Institute of Chemical Processes, Seoul National University, Seoul 08826, Republic of Korea. ✉email: thyeon@snu.ac.kr; s.wippermann@mpie.de; gerd.bacher@uni-due.de

The tunability of the fundamental bandgap, $E_g$, through size control on the nanometer scale is a key attribute of semiconductor nanocrystals, enabling applications[1,2] spanning from biomedical imaging[3] to optoelectronic devices such as light emitting diodes[4,5] and photodetectors[6]. The bandgap of semiconductors with reduced dimensions can usually be derived from the bulk value, correcting for electronic perturbations[7,8], such as charge carrier quantization and Coulomb interaction. Although the electronic structure of semiconductor nanocrystals is changed by quantum confinement, the temperature dependence of the bandgap usually appears to be dominated by bulk mechanisms in both epitaxial[9,10] as well as most colloidal quantum dots (QDs)[11–16]. However, there are few examples[17–19] that reported conflicting results on the temperature dependence of the bandgap in quantum dots, opening room for debate. For bulk, it is widely accepted that the temperature dependence of the bandgap results from a superposition of electron–phonon interactions and thermal lattice expansion[20,21]. Besides widely used empirical equations[20,22], the AHC approach[23–25] (after Allen, Heine and Cardona), which is based on second-order perturbation theory, approaches the temperature-induced bandgap shift as the change in energy levels when phonons are thermally populated. This has been used to obtain the temperature dependent bandgap computationally for a number of bulk semiconductors[26]. Following semi-empirical approaches, AHC was subsequently combined with density functional theory (DFT) or density functional perturbation theory (DFPT)[27–30]. For nanocrystals, the temperature dependence of the bandgap has been derived either (i) as time average of the bandgap with first principle molecular dynamic (MD) calculations[31–33], or (ii) via the frozen-phonon approach[34–36], summing over the Bose–Einstein weighted changes in eigenenergies due to atomic displacements along the different phonon modes.

Here, we demonstrate that the impact of temperature on the optical bandgap is fundamentally altered in materials at the border between solids and molecules. In magic sized clusters[37–40] (MSC) containing a defined number of 26 atoms, a pronounced exciton-induced weakening of the crystal bonds occurs, resulting in a giant shift of the phonon spectrum driven by the exciton. As a consequence, a drastic increase of $dE_g/dT$ (up to a factor of 2) with respect to bulk material or nanocrystals of typical size is found. Using a first principles approach, we computed the exciton-induced change of the phonon density of states from molecular dynamics simulations on the excited state potential energy surface. This enabled us to quantitatively describe our experimental data by deriving the bandgap shift with temperature as the vibrational entropy contribution to the free energy difference between the ground and optically excited states.

## Results

Our study benefits from the spectrally narrow absorption and emission resonances of high quality II–VI MSCs that consist of a well-defined number of atoms and vanishing size distribution. This allows us to trace the bandgap energy shift between 5 and 300 K with various optical techniques, as summarized in Fig. 1 for $(CdSe)_{13}$ MSCs. The absorption of $(CdSe)_{13}$ MSCs, which is dominated by multiple resonances affiliated to the fine structure of the lowest excitonic transition[41], exhibits a redshift with increasing temperature for the whole collectivity of peaks of about 160 meV (Fig. 1a) between cryogenic and room temperature. The short-time photoluminescence (PL) from the bandgap states (Fig. 1b), which is obtained by integrating the emission within the first 20 ps after laser excitation, is found to coincide with the low energy absorption feature of the band edge fine structure, with virtually no Stokes shift within our resolution (compare Supplementary Fig. 1b). This is in distinct contrast to what is observed

in typical QDs with Stokes shifts of up to 100 meV for diameters below 2 nm[42], and enables us to trace the temperature dependence of an individual state (Fig. 1b). In agreement with absorption, an energy shift of 160 meV is observed between cryogenic and room temperature.

Transition metal doping with manganese $(Mn^{2+})$[41] offers additional possibilities to track the bandgap shift for specific fine structure states. The sp-d exchange interactions, which persist up to room temperature in these clusters[43], generate a pronounced magneto-optical response at the bandgap, representing a powerful tool to distinguish between magneto-optically active and inactive states via magnetic circular dichroism (MCD) spectroscopy (Fig. 1c)[41]. For MSCs with $x_{Mn} = 2\%$ (average $Mn^{2+}$ content among the cations in all clusters of the measured ensemble), the shift of the bandgap slightly exceeds that of the undoped clusters (Fig. 1c), while the mean energy shift between cryogenic and room temperature among a concentration series ($x_{Mn} = 2$–10%, average out of 5 data sets) accounts for $164 \pm 16$ meV (Supplementary Table 1).

In addition, $Mn^{2+}$ ions modify the emission of the MSCs, leading to a characteristic orange luminescence originated in the internal $Mn^{2+}$ $^4T_1 \rightarrow {}^6A_1$ transition. Monitoring this emission in photoluminescence excitation (PLE) spectroscopy reveals a shift of about 200 meV of the bandgap absorption between 5 and 300 K (see Fig. 1d, $174 \pm 23$ meV in average among three doped samples). $Mn^{2+}$-doped $(ZnSe)_{13}$ MSCs[44] prepared for comparison exhibit a similarly enhanced temperature dependence as the CdSe based MSCs (Supplementary Fig. 2). As a reference, we synthesized CdSe QDs covered with amine ligands or a ZnSe shell, which exhibit significantly smaller bandgap shifts with temperature in this regime (average shift of both samples is $93 \pm 1$ meV in agreement with literature, see Fig. 2 and Supplementary Fig. 3). This demonstrates that our findings are related to the small size of the clusters, irrespective of the material system.

The bandgap shift with temperature as derived by different techniques for various doped and undoped MSCs are summed up in Fig. 2. The entity of evaluated data in the MSCs exhibits drastically enhanced values of $dE_g/dT$ in the high temperature regime as compared to typical CdSe QDs or the most common bulk semiconductors (ref. [21] and references therein). For $(CdSe)_{13}$ MSCs, our data reveals a mean increase of 89 % compared to bulk CdSe (wurtzite) for the high temperature limit of $dE_g/dT$ (between 150 and 300 K) among 14 data sets (Supplementary Table 1).

To reconstruct the impact of electron–phonon coupling on the temperature dependent bandgap shift, it is instructive to interpret the bandgap energy as the standard Gibbs free energy for the formation of an electron-hole pair[45]. As first suggested in the context of Brook's theorem[46], the bandgap change with temperature due to electron–phonon interactions can be expressed as the change in the vibronic free energy caused by the excitation of an electron from a bonding state in the valence band into a nonbonding state in the conduction band[24,47]. The limited number of atoms in our material allows us to directly calculate the free energy difference for $(CdSe)_{13}$ MSCs between the ground and the optically excited states as a function of temperature using constrained DFT. We performed the same set of calculations for the two most prominent structural models for the $(CdSe)_{13}$ MSCs[48]: the so-called sliced-wurtzite MSC without ligands (see Supplementary Fig. 4) and a core/cage MSC with methylamine ligands (Fig. 3a). In harmonic approximation, the free energy of the phonons is given by

$$F_{vib}(T) = \int d\omega\, g(\omega, T) \left( \frac{1}{2}\hbar\omega + k_B T \ln[1 - e^{-(\frac{\hbar\omega}{k_B T})}] \right), \quad (1)$$

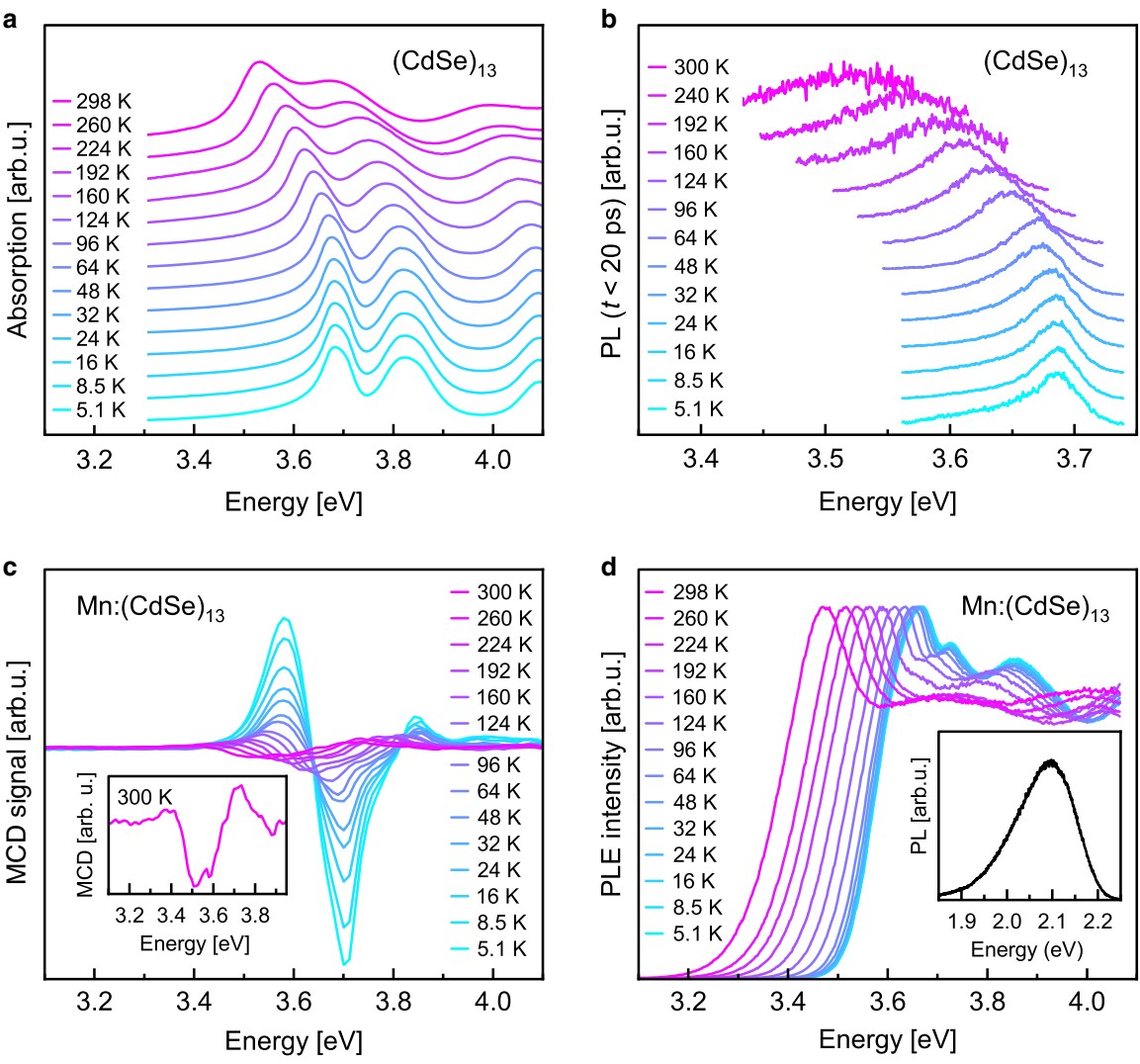

**Fig. 1 Temperature dependent bandgap shift in undoped as well as doped CdSe magic sized cluster (MSC). a** Absorption and **b** short-time photoluminescence (PL), measured between 5 K and 300 K. **c, d** Temperature dependence of the bandgap in Mn²⁺-doped (CdSe)₁₃ MSCs using optical signatures introduced by the magnetic dopants. **c** Magnetic circular dichroism (MCD) spectra of Mn:(CdSe)₁₃ MSCs at 1.6 T between 5.2 and 300 K. The signals allow tracking of the bandgap shift with temperature by monitoring the energetic position of the first zero-crossing. d, Photoluminescence excitation (PLE) spectra probed at the $^4T_1 \rightarrow {}^6A_1$ Mn²⁺ internal emission between 5.2 and 300 K. The inset depicts the time integrated PL signal of Mn²⁺-doped (CdSe)₁₃ MSCs revealing the specific orange Mn²⁺ luminescence, used for probing bandgap states in PLE experiments.

where $g(\omega, T)$ is the phonon density of states (PDOS). The first optically bright transition in the (CdSe)₁₃ MSC is the HOMO→LUMO+1 transition (highest occupied/lowest unoccupied molecular orbital, note that the LUMO describes a midgap state, which is related to the surface and frequently observed in CdSe nanoclusters[49–52], see Supplementary Fig. 6).

To obtain the excited state PDOS, we constrained the electron occupations of the orbitals, so that one electron from the HOMO is transferred to the LUMO+1 state. We calculated the phonon frequencies, $\omega_i$, for the cluster in the ground and the excited state from the force constants. Compared to the MSC in the ground state electronic configuration, the presence of the excited electron weakens the bonds and causes a red-shift of the average phonon frequency of 4.2% for the sliced-wurtzite structure (116.0 cm⁻¹→111.1 cm⁻¹) (individual phonon frequencies in Supplementary Tables 2 and 3). This shift in the mean frequency is almost two orders of magnitude larger than calculated for bulk Si (<0.1% at the melting temperature)[24]. Inserting the obtained phonon frequencies, the term for the free

energy is reduced to a summation. Neglecting the explicit temperature and volume dependence of the PDOS ($g(\omega, T) \approx g(\omega, T = 0\,\mathrm{K})$), we obtain the bandgap shift with temperature

$$\Delta E_g(T) = E_g(T) - E_g(0\,\mathrm{K}) \tag{2}$$

as the free energy difference

$$\Delta E_g(T) = F_{\mathrm{vib,exc}}(T) - F_{\mathrm{vib,gs}}(T) - \left( F_{\mathrm{vib,exc}}(0\,\mathrm{K}) - F_{\mathrm{vib,gs}}(0\,\mathrm{K}) \right) \tag{3}$$

between the ground and excited state electronic configurations (blue lines in Fig. 3c). For $T < 150\,\mathrm{K}$, the measured and calculated temperature-dependence of $\Delta E_g$ agree very closely. At higher temperatures, the experimental data deviate slightly from a perfectly linear behavior, indicating anharmonic contributions to the free energy.

In order to explicitly account for the temperature-dependence of the phonon frequencies, we used the same constrained DFT approach to perform ab initio MD calculations to obtain the

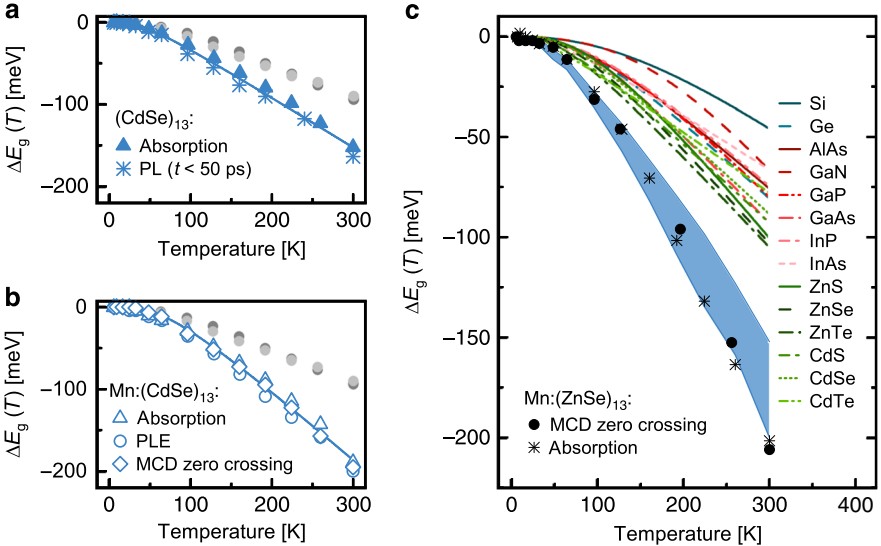

**Fig. 2 Giant temperature dependent shift of the bandgap in magic sized cluster (MSC). a, b** Comparison of the shift in transition energies from cryogenic to room temperature ($\Delta E_g$) for undoped (**a**) and $Mn^{2+}$-doped $(CdSe)_{13}$ MSCs (**b**) as extracted from different experimental approaches. Data for conventional CdSe (light grey) and CdSe/ZnS (dark grey) quantum dots (QDs) are displayed as dots for comparison (see Supplementary Fig. 3 for the spectra). The peak positions for absorption, photoluminescence (PL) and photoluminescence excitation spectroscopy (PLE) signals are extracted via Gaussian fits from the spectra. The peak positions for the energetically lowest magneto-optically active transition are taken from MCD zero-crossings (Fig. 1c). Solid lines represent guides to the eye. **c** Comparison of the temperature dependence of the band edge transitions for the entity of $(CdSe)_{13}$ (blue area) and $(ZnSe)_{13}$ MSCs (black symbols) analyzed within this study to some of the most important group IV, III–V, and II–VI semiconductors (data sets are plotted using parametrization following ref. [21]). The bandgap shifts for $Mn^{2+}$-doped $(ZnSe)_{13}$ MSCs are extracted from MCD zero-crossings (black circles) and absorption (black stars) (spectra shown in Supplementary Fig. 2).

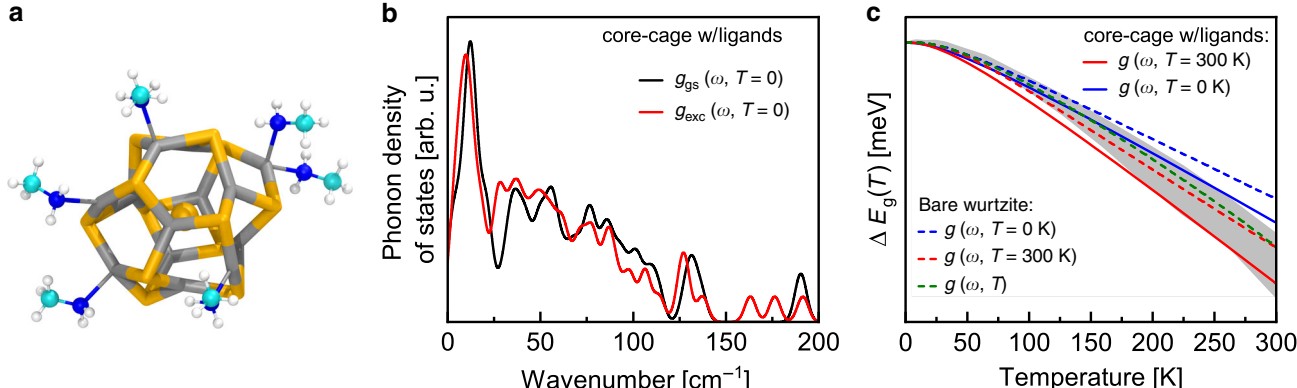

**Fig. 3 Calculated phonon density of states (PDOS) and bandgap relaxation with temperature. a** Relaxed structural model of a core/cage $(CdSe)_{13}$ magic sized cluster (MSC) with methylamine (MA) ligands in the ground state electronic configuration. **b** Normalized PDOS for core/cage $(CdSe)_{13}$ MSC as shown in **a** in the ground and HOMO→LUMO+1 excited state electronic configurations, respectively, used to calculate $\Delta E_g(T, g_{corecage}(\omega, T = 0 K))$ (blue straight line in **c**). Note, that beside the modes inherent to the CdSe cluster, the PDOS also contains vibrational and rotational modes stemming from the MA ligands. **c** Temperature-dependent bandgap shift of $(CdSe)_{13}$ MSCs calculated using phonon frequencies obtained in harmonic approximation $g(\omega, T = 0 K)$ (blue), including anharmonicities $g(\omega, T = 300 K)$ (red) and with full temperature dependence $g(\omega, T)$ (green), in comparison to experiment (grey area). Solid and dashed lines denote the core/cage MSC with MA ligands and the sliced-wurtzite MSC without ligands, respectively.

PDOS at 300 K (see Methods). Inserting the ground and excited state PDOSs for $T = 300 K$, we obtained a slight overestimation for the temperature-dependence of the bandgap (red lines in Fig. 3c). In the case of the sliced-wurtzite $(CdSe)_{13}$ MSCs, we then approximated the temperature-dependent PDOS $g(\omega, T)$ using Gaussian functions as a basis set and interpolated between the PDOSs at $T = 0 K$ and $T = 300 K$ (see "Methods"). As shown in Fig. 3c, the bandgap calculated ab initio from the fully temperature-dependent $g(\omega, T)$ (green dashed line) is in excellent agreement with the experiments within the whole measured temperature range.

## Discussion

We now discuss our results in connection to current literature. Previous DFT calculations for nanoclusters determined the temperature dependent bandgap either (i) as the mean HOMO-LUMO energy difference in a MD[31–33], which distributes the atomic velocities according to classical Maxwell-Boltzmann statistics instead of Bose–Einstein occupation of the vibrational modes, and thus inherently overestimates $dE_g/dT$ at low temperatures, or (ii) by summing over Bose–Einstein weighted eigenenergy changes induced by frozen-phonon modes[34–36]. The latter is known to severely overestimate the temperature dependence of the bandgap

in nanocrystals[35,36], often attributed to overrated contributions of surface and ligand modes. Indeed, we obtain an overestimation by almost an order of magnitude, calculating the bandgap shift with temperature based on a frozen-phonon approach (compare Supplementary Fig. 5). The failure of the commonly used frozen-phonon approach, which is very successful in the bulk, indicates that there is something fundamentally different about the temperature-dependence of the band gap in NCs. The approach presented in this work, which explicitly incorporates the effect of photon absorption on phonon modes, is able to successfully describe $dE_g/dT$ over the entire temperature range between 5 and 300 K with quantitative accuracy and without any free parameters. Brook's theorem[46] states that the temperature-dependence of the bandgap can be obtained equally from either the changes in the eigenenergies induced by the phonons or the changes in the vibrational frequencies induced by the optical excitations. Clearly, our results demonstrate that Brook's theorem is invalid for small nanoclusters and they highlight the importance of taking excitonic effects into account explicitly.

For high temperatures $(\hbar\omega \ll k_B T)$, the slope of the temperature-induced band gap shift can be estimated as

$$\frac{dE_g}{dT} \propto \ln\left(\frac{\omega_{exc}}{\omega_{gs}}\right) = \ln\left(1 + \frac{\Delta\omega}{\omega_{gs}}\right), \quad (4)$$

disclosing two factors which might enhance $dE_g/dT$ in nanoclusters compared to bulk: (i) Softer phonon modes in general compared to bulk, e.g. due to surface related acoustic modes, and (ii) an enhanced change $\Delta\omega$ in the phonon energies due to the absorption process. The former has been predicted for PbS[53] and Si[35], while for CdSe nanoparticles theoretically[36,54,55] and experimentally[54] a blueshift of the acoustic surface modes has been observed. Our calculations reveal a distinct redshift in the acoustic modes for the core-cage nanocluster model compared to bulk, while the data exhibit no clear shift for the sliced-wurtzite structure (see Supplementary Figure 4c). Both structures reveal enhanced temperature-induced bandgap shifts (red lines in Fig. 3c), indicating that the main origin lies in the significant change in the PDOS of clusters in the presence of an exciton. Due to the small size of the clusters, the exciton is distributed over a minimal number of 40–50 bonds. Note that one exciton per cluster corresponds to a photo-generated charge carrier density in the range of $10^{21}$ cm$^{-3}$, representing the regime of particularly high optical pumping in conventional semiconductors.

Our results not only highlight how the absorption of one photon as the probe medium influences the observed bandgap, but also visibly demonstrate how the temperature dependence of the bandgap immediately depends on the phonon dispersion of a material. We expect these findings to be relevant in materials that are either quantum-confined, strongly correlated or exhibit strong electron–phonon coupling and thus optical excitations significantly alter the vibrational free energies and in turn derived properties.

## Methods

**Sample preparation.** Nanocrystals and MSCs used in this study were synthesized using standard Schlenk techniques under an argon atmosphere. The MSCs were obtained from a Lewis acid-base reaction between metal-ammine halide complexes ($CdCl_2(octylamine)_2$, $ZnCl_2(octylamine)_2$, and $MnCl_2(octylamine)_2$) and octyl-ammonium selenocarbamate (0.67 M) in $n$-octylamine (Aldrich, 99%). For $Mn^{2+}$:($CdSe)_{13}$ MSCs, 1.0 mmol of $CdCl_2$ and 0.1 mmol of $MnCl_2$ were heated in 7 ml of n-octylamine at 120 °C for 2 h. The average doping concentration is changed by adjusting the initial precursor ratio ($MnCl_2/CdCl_2$). Octylammonium seleno-carbamate was prepared by bubbling CO gas into 3 ml of $n$-octylamine containing 2.0 mmol Se powder for 1 h at room temperature. This solution was injected into the as-prepared precursor mixture containing metal-ammine halide complexes and kept at room temperature for more than 24 h. The MSCs were precipitated by adding ethanol containing trioctylphosphine followed by washing several times with ethanol[1–3]. For the synthesis of $Mn^{2+}$:($ZnSe)_{13}$ MSCs, $ZnCl_2$ was used instead of $CdCl_2$. The average doping concentration of the MSCs was measured by

inductively coupled plasma-atomic emission spectrometry (ICP-AES, Shimadzu ICPS-7500). To enhance the dispersibility of MSCs, which results in minimization of light scattering, the initial octylamine ligands were exchanged with long-chain oleylamine ligands[1]. CdSe QDs and CdSe/ZnS QDs were synthesized by conventional hot injection method using Cd(oleate)$_2$ and trioctylphosphine selenide (TOPSe) as precursors[4,56]. In all, 1.2 mmol CdO were added to a mixture of 1.5 mL of oleic acid and 20 mL of 1-octadecene and heated under vacuum for 2 h at 120 °C to form Cd(oleate)$_2$ complexes. The mixture was then heated to 300 °C under Ar and TOPSe (1 M) was rapidly injected to obtain CdSe QDs. For the growth of ZnS shells, 4.8 mmol of Zn(oleate)$_2$ and 4.8 mmol of tributylphosphine sulfide were additionally introduced to the as-prepared solution containing CdSe QDs and the reaction mixture was kept at 300 °C for 20 min. For CdSe QDs and CdSe/ZnS QDs, we also performed a surface modification with oleylamine to neglect the effect of the surface ligands.

**Optical characterization.** For absorption, photoluminescence excitation spectroscopy (PLE) and magnetic circular dichroism (MCD) measurements the MSCs passivated by oleylamine were prepared as thin films between two quartz glass substrates, and for PL measurements between a silicon and a quartz glass substrate. MCD spectroscopy was conducted on a homemade setup consisting of a 75 W Xenon lamp (Lot-Oriel) equipped with a monochromator (omni-λ 150, Lot-Oriel) and a photomultiplier (R928, Hamamatsu). The excitation light was modulated using a photoelastic modulator (PEM-90, Hinds Instruments). The sample was placed in a helium vapor cryostat (ST-300, Janis) between two poles of an electromagnet (EM4-HVA, Lake Shore) in Faraday geometry. Absorption measurements were either extracted from the DC signal during MCD measurements or collected with an UV-VIS spectrophotometer (Shimadzu 2550) equipped with a helium vapor cryostat (ST-300, Janis). PLE measurements were conducted with a helium vapor cryostat (ST-300, Janis) placed in a Fluorolog-3 (FL3-22, Horiba Scientific). For time resolved analysis of the PL signal the samples placed in a helium vapor cryostat (ST-300, Janis) were excited with a frequency tripled Ti: sapphire laser (Mira 900, Coherent, 270 nm) pumped with an Nd:YAG laser (Verdi-V10, Coherent) with 100 fs pulses at a repetition rate of 76 MHz. The signal was detected with a streak camera system consisting of a monochromator (250IS, Bruker Optics) and a Synchroscan Streak camera (C5680-24 c, Hamamatsu), providing a temporal resolution of 4 ps.

**DFT calculations.** We carried out first principles calculations with density functional theory and plane-wave basis sets, within the Perdew-Burke-Ernzerhof (PBE)[57,58] approximation as implemented in the Quantum Espresso package[59]. We used norm-conserving pseudopotentials and a wavefunction energy cutoff of 45 Ry. The molecular dynamics (MD) calculations were performed at 300 K, using a time step of 2 fs and the Berendsen[60] thermostat with a rise time of 10 time steps. The clusters were equilibrated for 25 ps and subsequently data was sampled for 100 ps. Excited states were described by pair excitations[61] within constrained DFT, where the Kohn-Sham energy is minimized under the constraint that the occupation of the highest occupied molecular orbital (HOMO) is reduced by 1 e$^-$, compared to the ground state, and the occupation of the lowest energy state directly above the lowest unoccupied molecular orbital (LUMO + 1) is increased by 1 e$^-$. To prevent convergence issues within the electronic loop due to level crossings and concurrent discontinuous changes in the charge density, we implemented a smearing scheme into Quantum Espresso, where a Gaussian smearing is applied individually to the electron and the hole with a smearing of $\sigma = 0.001$ Ry. At $T = 0$ K, phonon eigenfrequencies and eigenvectors are calculated from finite differences. At $T = 300$ K, phonon densities of states (PDOS) are computed from the MD trajectories using the maximum entropy method (MEM)[62,63], the same numerical setup as above and 4096 poles in the frequency interval from 0 cm$^{-1}$ to 300 cm$^{-1}$. The spectra are renormalized so that the integrals $\int d\omega g(\omega)$ are equal to the number of phonon modes. To calculate $g(\omega, T)$, we approximate the set of discrete phonon frequencies for $T = 0$ K by Gaussians with a very small but finite width $\sigma = 2$ cm$^{-1}$ according to:

$$g(\omega, T = 0) = \Sigma_i \frac{1}{\sqrt{2\pi\sigma^2}} e^{-\frac{(\omega - \omega_i)^2}{2\sigma^2}} \quad (5)$$

Subsequently, $g(\omega, T = 300\,K)$ is expanded into the same number of Gaussians, albeit with modified positions and widths. All resulting PDOS within the Gaussian basis set have been confirmed to yield the same free energies as before. We then linearly interpolate the widths and positions of the Gaussians as a function of temperature to obtain $g(\omega, T)$.

## Data availability

The source data underlying Figs. 1–3b, c) are provided in a source data file. All other relevant data are available from G.B. on request. Source data are provided with this paper.

## Code availability

The computer code used for the DFT and MD calculations during the study is available from S.W. on request.

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

## Acknowledgements

F.M., S.L., and G.B. gratefully acknowledge financial support from the German Research Foundation DFG under contract Ba 1422/13-2. F.M. in addition was supported by the German Academic Exchange Service (DAAD) with funds from the German Federal Ministry of Education and Research (BMBF) and the European Union (FP7-PEOPLE-2013-COFUND - grant agreement n° 605728). T.A.N., E.S., and S.W. were supported by the German Federal Ministry of Education and Research (BMBF) within the Nano-MatFutur programme, grant no. 13N12972. Supercomputer time provided by NERSC (project no. 35687) and the Max-Planck Computing and Data Facility, Garching, is gratefully acknowledged. T.H. acknowledges financial support by the Research Center Program of the Institute for Basic Science (IBS) in Korea (IBS-R006-D1). J.Y. acknowledges the financial support by the DGIST Start-up Fund Program of the Ministry of Science, ICT and Future Planning (2019070014).

## Author contributions

F.M. and G.B. conceived and designed the experiments, led the research and wrote the paper. J.Y. and T.H. designed and prepared the materials. S.L. contributed to the optical measurements, data analysis and interpretation. S.W. conceived and designed the DFT calculations, T.A.N. and E.S. conducted the DFT calculations. All authors discussed the results and assisted in paper preparation.

## Competing interests

The authors declare no competing interests.
