## [Peer Review File · Nature Communications]

Reviewers' comments:

Reviewer #1 (Remarks to the Author):

Referee report on the manuscript "Exciton-driven shift of phonon modes causes giant temperature dependence of the bandgap in nanoclusters" by Muckel et al.

The authors measure and calculate the temperature dependence of the band gap of magic sized CdSe nanoclusters. They find a strong temperature dependence, increased by a factor of two compared to bulk. They compare their experimental results with theory and obtain excellent agreement. They conclude that the strong increased temperature dependence originates from an excitonic effect.

The manuscript is generally well written and kept rather short and focussed, which is appropriate for the journal. I have the impression that the experimental work is solid and the theory is defensible, although a few questions arise summarized below. I enjoyed reading the manuscript and I truly appreciate the topic which is simple and fundamental: temperature dependence of the band gap.

My main concern with the work is about the conclusions drawn and the big claims made. A very small CdSe NC has a stronger (by a factor of 2) temperature dependence than bulk CdSe. Which is as such not astonishing, at which point the word "giant" claimed by the authors may be questioned. The more important and more intricate point is about the excitonic origin of the effect. I would challenge this statement as well. If I would calculate the temperature dependent band gap in these small clusters using an approach that uses no exciton. Using for instance the AHC approach or a frozen phonon method, i.e., the change of the electronic eigenvalues resulting from the phonon (instead of the change of frequency resulting from the changed electronic occupation, as the authors do), I would presumably get the same results, if we all believe in Brooks' theorem which makes this connection. Now Brooks theorem is for bulk and the authors use it for a very small NC and change the occupation drastically, which may seem in contradiction to the original Brooks work. I could be convinced by the importance of the excitonic effect if the results of AHC or frozen phonon would be significantly different (and in disagreement with experiment) than the results of this work (in agreement with experiment). This would be very important and fundamental and would deserve a Nature Communication publication. At the moment I cannot see a convincing evidence for it.

A few more technical points:

- 1) The title should reflect that the results are for CdSe and not for "nanoclusters".
- 2) Line 45, in which temperature range?
- 3) Line 59, how can you have 2% Mn in a 26 Atom NC?
- 4) Is it maybe interesting to talk about the fact that the HOMO-LUMO transition is forbidden? Or is this known already?
- 5) Line 133. Please add a short note why MD overestimates the occupation of higher energy phonon modes.
- 6) Why is in Fig.3b the phonon DOS starting at 0? For such a small NC, there is a phonon gap and no very low frequency phonons. Also in the supplementary. It looks to me as there are some problems with the phonon-DOS?

Reviewer #2 (Remarks to the Author):

In the manuscript presented here, Bacher and coworkers investigate the temperature dependent

bandgap of (CdSe)₁₃ clusters.

Combining theoretical and experimental approaches, they show the importance of phonon modes on the temperature dependent bandgap of such magic sized clusters. The conclusions are sound and the calculations are novel, and thus, this work will be impactful to the scientific community.

The manuscript is well written, and the conclusions are well supported by data and theory, and I believe the manuscript can be published with only minor changes. The manuscript could benefit from a brief introduction into the field, at present the authors dive right into the field and it may be beneficial to have an overview of what has been done in this direction previously.

Answer to the Reviewer's Comments

Reviewer #1 (Remarks to the Author):

Referee report on the manuscript "Exciton-driven shift of phonon modes causes giant temperature dependence of the bandgap in nanoclusters" by Muckel et al.

The authors measure and calculate the temperature dependence of the band gap of magic sized CdSe nanoclusters. They find a strong temperature dependence, increased by a factor of two compared to bulk. They compare their experimental results with theory and obtain excellent agreement. They conclude that the strong increased temperature dependence originates from an excitonic effect.

The manuscript is generally well written and kept rather short and focussed, which is appropriate for the journal. I have the impression that the experimental work is solid and the theory is defensible, although a few questions arise summarized below. I enjoyed reading the manuscript and I truly appreciate the topic which is simple and fundamental: temperature dependence of the band gap.

Our Answer:

We thank the reviewer for their encouraging words about the scope of the paper and the appreciation of the scientific quality.

Comment #1:

My main concern with the work is about the conclusions drawn and the big claims made. A very small CdSe NC has a stronger (by a factor of 2) temperature dependence than bulk CdSe. Which is as such not astonishing, at which point the word "giant" claimed by the authors may be questioned. The more important and more intricate point is about the excitonic origin of the effect. I would challenge this statement as well. If I would calculate the temperature dependent band gap in these small clusters using an approach that uses no exciton. Using for instance the AHC approach or a frozen phonon method, i.e., the change of the electronic eigenvalues resulting from the phonon (instead of the change of frequency resulting from the changed electronic occupation, as the authors do), I would presumably get the same results, if we all believe in Brooks' theorem which makes this connection. Now Brooks' theorem is for bulk and the authors use it for a very small NC and change the occupation drastically, which may seem in contradiction to the original Brooks work. I could be convinced by the importance of the excitonic effect if the results of AHC or frozen phonon would be significantly different (and in disagreement with experiment) than the results of this work (in agreement with experiment). This would be very important and fundamental and would deserve a Nature Communication publication. At the moment I cannot see a convincing evidence for it.

Our answer:

The referee raises a very important point here. Accurately calculating the temperature-dependent bandgap of NCs is a longstanding problem. We followed the suggestion of the referee and calculated the bandgap using the frozen phonon method from the change of the electronic eigenvalues.

Figure 1 Comparison of the bandgap shift with temperature, calculated using the frozen phonon approach (red) and the change in vibrational free energy (blue), both in harmonic approximation. The range of experimental data is highlighted as a light blue area. The renormalization energy calculated via the frozen phonon approach is 397 meV, while it accounts for 56 meV if calculated using the change in vibrational free energy for an exciton.

Figure 1 compares the frozen phonon results (red curve) to our exciton free energy approach (blue curve), both in harmonic approximation. The ground state phonons, from which we obtained the red curve, are the same ones that enter the exciton free energy calculation. By disregarding the excitonic effect, we obtain a result that is wrong by almost an order of magnitude!

In fact, frozen phonon's severe overestimation of the temperature-dependent bandgap in small NCs has long been known: these results are in line with previous findings, see e. g., ref. 1, Fig. 2a. This overestimation is often attributed to the presence of acoustic surface or ligand modes, which are then removed manually from the calculation to obtain a reasonable agreement, cf. ref. 1, Fig. 2c. However, a pure first principles calculation of the temperature-dependent band gap in NCs has so far remained elusive. Here, the complete failure of the commonly used frozen phonon approach, which is very successful in the bulk, indicates that there was something fundamentally wrong with our understanding of the temperature-dependence of the band gap in NCs. Brook's theorem states that the equality between obtaining the temperature-dependence of the band gap either from changes in the eigenenergies induced by the phonons or the changes in the vibrational frequencies induced by the optical excitations. Clearly, our results demonstrate that Brook's theorem is invalid for small nanoclusters and they highlight the importance of taking excitonic effects into account explicitly.

The advance in our paper is two-fold: (1) measurements for magic-sized NCs, that have no spread in size or structural details, allow us to measure the temperature-dependent bandgap with extremely small error bars. (2) With our proposed exciton free energy approach, we have for the first time a pure ab initio model that quantitatively agrees with experiment and has no free parameters. We expect that our results will trigger new research and finally allow us to understand the exact mechanistic details of the temperature dependence of the bandgap in NCs.

To account for the reviewer's concerns, we modified the discussion of our results in context with current literature, taking into account the failure of the frozen phonon approach to describe our experimental data. We also substitute the word "giant" by "strong" in the title.

Changes in main manuscript and SI:

"Exciton-driven shift of phonon modes causes strong temperature dependence of the bandgap in II-VI nanoclusters"

Changes in the main manuscript:

“We now discuss our results in connection to current literature. Previous DFT calculations for nanoclusters determined the temperature dependent bandgap either (1) as the mean HOMO-LUMO energy difference in a MD²⁻⁴, which distributes the atomic velocities according to classical Maxwell-Boltzmann statistics instead of Bose-Einstein occupation of the vibrational modes, and thus inherently overestimates $\frac{dE_g}{dT}$ at low temperatures, or (2) by summing over Bose-Einstein weighted eigenenergy changes induced by frozen-phonon modes^{1,5,6}. The latter is known to severely overestimate the temperature dependence of the bandgap in nanocrystals^{1,6}, often attributed to overrated contributions of surface and ligand modes. Indeed, we obtain an overestimation by almost an order of magnitude, calculating the bandgap shift with temperature based on a frozen-phonon approach (compare Supplementary Figure 5). The failure of the commonly used frozen-phonon approach, which is very successful in the bulk, indicates that there is something fundamentally different about the temperature-dependence of the band gap in NCs. The approach presented in this work, which explicitly incorporates the effect of photon absorption on phonon modes, is able to successfully describe $\frac{dE_g}{dT}$ over the entire temperature range between 5 K and 300 K with quantitative accuracy and without any free parameters. Brook’s theorem⁷ states that the temperature-dependence of the bandgap can be obtained equally from either the changes in the eigenenergies induced by the phonons or the changes in the vibrational frequencies induced by the optical excitations. Clearly, our results demonstrate that Brook’s theorem is invalid for small nanoclusters and they highlight the importance of taking excitonic effects into account explicitly. “

Changes in the SI:

Supplementary Figure 5 | Frozen Phonon approach. Comparison of the bandgap shift with temperature, calculated using the frozen phonon approach (red) and the change in vibrational free energy (blue), both in harmonic approximation. The ground state phonons at 0 K (compare grey trace in Supplementary Figure 4b)), which enter the exciton free energy calculation, are the same ones that are used in the frozen-phonon approach. The range of experimental data is highlighted as a light blue area. The zero point motion renormalization energy calculated via the frozen phonon approach is 397 meV, while it accounts for 56 meV if calculated using the change in vibrational free energy for an exciton.”

A few more technical points:

Comment #2:

1) The title should reflect that the results are for CdSe and not for “nanoclusters”.

Our answer:

As we observe basically the same enhanced temperature shift in ZnSe nanoclusters, we believe that our results are generally applicable to II-VI nanoclusters. To clarify the material class, we thus added the term “II-VI” to the title in both, the main manuscript and SI.

Comment #3:

2) Line 45, in which temperature range?

Our answer:

The shift in energy refers to the range between 5 K and 300 K, as given in the sentence above.

We added a reference to this temperature range in line 45 of the main manuscript:

“The absorption of (CdSe)₁₃ MSCs, which is dominated by multiple resonances affiliated to the fine structure of the lowest excitonic transition⁸, exhibits a redshift with increasing temperature for the whole collectivity of peaks of about 160 meV (Fig. 1a) **between cryogenic and room temperature.**”

Comment #4:

3) Line 59, how can you have 2% Mn in a 26 Atom NC?

Our answer:

We are sorry that we failed to word more precisely here. We achieve an average of 2% Mn concentration among the entity of cations in all clusters of the ensemble measured. With the dopants being randomly distributed, we expect a mixture of clusters containing non, one or two Mn ions (see Muckel et al., ACS Nano 10, 7135, 2016). We altered the wording to make this clear:

“For MSCs with $x_{\text{Mn}}=2\%$ **(average Mn²⁺ content among the cations in all clusters of the measured ensemble)**, the shift of the bandgap slightly exceeds that of the undoped clusters (Fig. 1c), while the mean energy shift between cryogenic and room temperature among a concentration series ($x_{\text{Mn}}=2\%$ to 10%) accounts for 164±16 meV (Supplementary Table 1).”

Comment #5:

*4) Is it maybe interesting to talk about the fact that the HOMO-LUMO transition is forbidden?
Or is this known already?*

Our answer:

We thank the reviewer for pointing out this aspect. This is definitely of interest and not yet well understood. It has been reported (e.g. ⁹⁻¹²) that in small CdSe nanoclusters the surface tends to reorganize in a way that a mid-gap state is formed (referred to as LUMO in our work), although the exact origin has never been resolved satisfactorily. This state is observed to be weak in absorption, but more prominent in photoluminescence⁹⁻¹², depended on the temperature and quality of the sample. Enclosed you find a 5 K full range, time integrated photoluminescence spectrum of our undoped clusters at slightly higher excitation power, showing an additional broad luminescence

around 3.2 eV, which we hypothesize to stem from the mid-gap state. We are actively working on this topic and will present the results in an upcoming publication. For this work here, we are able to separate the band edge emission (3.65 eV) from this mid-gap state related emission by monitoring the transient PL at lower excitation intensities, compare Supporting Figure 1.

Figure 2 Full Photoluminescence spectrum at 5 K, excited with the 270 nm Ti:Sapphire line at 30 μ W total intensity.

We include this discussion in the Supporting Information Figure 6, with a reference in the main manuscript.

Changes in the main manuscript (line 128):

“The first optically bright transition in the $(\text{CdSe})_{13}$ MSC is the HOMO \rightarrow LUMO+1 transition (note that the LUMO describes a mid-gap state, which is related to the surface and frequently observed in CdSe nanoclusters⁹⁻¹², see Supplementary Fig. 6) “

Changes in the SI:

“**Supplementary Figure 6 | Full time-integrated PL spectrum of undoped cluster at 5 K and an excitation power of 30 μ W.** Besides the band edge transition at 3.65 eV, the spectrum depicts a broad luminescence around 3.2 eV, which we hypothesize to result from the (HOMO \rightarrow LUMO) transition, with the LUMO describing a mid-gap state stemming from the surface. It has been frequently reported⁹⁻¹² that in small CdSe nanoclusters the surface tends to reorganize in a way that a mid-gap state is formed (referred to as LUMO in our work). This state is observed to be weakly absorbing, but more prominent in photoluminescence⁹⁻¹², depended on temperature and quality of the sample. However, at lower excitation intensities and monitoring the transient PL, we are able to separate the band edge emission around 3.65 eV from the mid-gap state related emission around 3.2 eV.”

Comment #6:

5) Line 133. Please add a short note why MD overestimates the occupation of higher energy phonon modes.

Our answer:

We thank the referee for pointing out this issue. Unless the simulation temperature T is significantly above the Debye temperature Θ , MD overestimates the occupation of lower energy phonon modes. First principles MD (usually) treats the atomic nuclei as classical particles. Hence, atomic velocities obey Maxwell-Boltzmann statistics. In the real physical system, however, the phonon modes are occupied according to Bose-Einstein statistics. For $T < \Theta$, Maxwell-Boltzmann statistics inherently overestimate in particular the occupation of low energy phonons.

We corrected the sentence and added a short note in the main manuscript (line 160):

“Previous DFT calculations for nanoclusters determined the temperature dependent bandgap either (1) as the mean HOMO-LUMO energy difference in a MD²⁻⁴, which distributes the atomic velocities according to classical Maxwell-Boltzmann statistics instead of Bose-Einstein occupation of the vibrational modes, and thus inherently overestimates $\frac{dE_g}{dT}$ at low temperatures, or (2) by summing over Bose-Einstein weighted eigenenergy changes induced by frozen-phonon modes^{1,5,6}.”

Comment #7:

6) Why is in Fig.3b the phonon DOS starting at 0? For such a small NC, there is a phonon gap and no very low frequency phonons. Also in the supplementary. It looks to me as there are some problems with the phonon-DOS?

Our answer:

Our answer:

We thank the reviewer for his attentive remark. We carefully checked again the phonon densities of states, and did not find any mistakes. Supplementary Fig. 4c shows the phonon DOS of bulk CdSe (upper panel) and a bare wurtzite (CdSe)₁₃ NC (middle panel), which has no low frequency phonons. Figure 3b as well as the bottom panel of Supplementary Figure 4c depict PDOSs calculated for a (CdSe)₁₃ core/cage cluster passivated with 6 Methylamine ligands. The cluster itself does not exhibit any low frequency phonons. Instead, the phonon DOS below 25 cm⁻¹ originates completely from vibrations of the MA ligands (mostly rotational modes, which have low force constants).

We added a note addressing this issue to the main manuscript as well as the SI.

Changes in the main manuscript, Figure caption of Figure 3:

“**b**, Normalized PDOS for core/cage (CdSe)₁₃ MSC as shown in **a** in the ground and HOMO→LUMO+1 excited state electronic configurations, respectively, used to calculate $\Delta E_g(T, g_{corecage}(\omega, T = 0 K))$ (blue straight line in **c**). **Note, that beside the modes inherent to the CdSe cluster, the PDOS also contains vibrational and rotational modes stemming from the MA ligands.**”

Changes in the SI, Figure caption of Figure 4:

“Supplementary Figure 4 | PDOS of the sliced-wurtzite (a) and core/cage (b) models at different temperatures. Note, that for the core/cage cluster with MA ligands the PDOS contains in addition to modes inherent to the CdSe cluster vibrational and rotational modes stemming from the MA ligands, which fill up the phonon gap towards zero energy. a, Crystal structure of the bare sliced-wurtzite MSC.”

Reviewer #2 (Remarks to the Author):

In the manuscript presented here, Bacher and coworkers investigate the temperature dependent bandgap of (CdSe)₁₃ clusters.

Combining theoretical and experimental approaches, they show the importance of phonon modes on the temperature dependent bandgap of such magic sized clusters. The conclusions are sound and the calculations are novel, and thus, this work will be impactful to the scientific community.

The manuscript is well written, and the conclusions are well supported by data and theory, and I believe the manuscript can be published with only minor changes.

Our answer:

We thank the reviewer for their positive statement about our work and appreciate the evaluation that the manuscript deserves publication with only minor changes.

Comment #1:

The manuscript could benefit from a brief introduction into the field, at present the authors dive right into the field and it may be beneficial to have an overview of what has been done in this direction previously.

Our answer:

We thank the reviewer for his suggestion. According to the style guideline for Nature Communications, we modified the structure of the paper. Now, it contains an unreferenced abstract and a referenced introduction, which includes an overview of what has been done previously, as requested by the reviewer.

Changes to the main manuscript:

Abstract:

The fundamental bandgap E_g of a semiconductor – often determined by means of optical spectroscopy - represents its characteristic fingerprint and changes distinctively with temperature. Here, we demonstrate that in magic sized II-VI clusters containing only 26 atoms, a pronounced weakening of the bonds occurs upon optical excitation, which results in a strong exciton-driven shift of the phonon spectrum. As a consequence, a drastic increase of $\frac{dE_g}{dT}$ (up to a factor of 2) with respect to bulk material or nanocrystals of typical size is found. We are able to describe our experimental data with excellent quantitative agreement from *first principles* deriving the bandgap shift with temperature as the vibrational entropy contribution to the free energy difference between the ground and optically excited states. Our work demonstrates, how in small

nanoparticles photons as the probe medium affect the bandgap - a fundamental semiconductor property.

Manuscript:

The tunability of the fundamental bandgap, E_g , through size control on the nanometer scale is a key attribute of semiconductor nanocrystals, enabling applications^{13,14} spanning from biomedical imaging¹⁵ to optoelectronic devices such as light emitting diodes^{16,17} and photodetectors¹⁸. The bandgap of semiconductors with reduced dimensions can usually be derived from the bulk value, correcting for electronic perturbations^{19,20} such as charge carrier quantization and Coulomb interaction. Although the electronic structure of semiconductor nanocrystals is changed by quantum confinement, the temperature dependence of the bandgap usually appears to be dominated by bulk mechanisms in both epitaxial^{21,22} as well as most colloidal quantum dots^{23–28}. However, there are few examples^{29–31} that reported conflicting results on the temperature dependence of the bandgap in quantum dots, opening room for debate. For bulk, it is widely accepted that the temperature dependence of the bandgap results from a superposition of electron-phonon interactions and thermal lattice expansion^{32,33}. Besides widely used empirical equations^{32,34}, the AHC approach^{35–37} (after Allen, Heine and Cardona), which is based on second-order perturbation theory, approaches the temperature-induced bandgap shift as the change in energy levels when phonons are thermally populated. This has been used to obtain the temperature dependent bandgap computationally for a number of bulk semiconductors³⁸. Following semi-empirical approaches, ACH was subsequently combined with density functional theory (DFT) or density functional perturbation theory (DFPT)^{39–42}. For nanocrystals, the temperature dependence of the bandgap has been derived either (1) as time average of the bandgap with first principle molecular dynamic (MD) calculations^{2–4}, or (2) via the frozen-phonon approach^{1,5,6}, summing over the Bose-Einstein weighted changes in eigenenergies due to atomic displacements along the different phonon modes.

Here, we demonstrate that the impact of temperature on the optical bandgap is fundamentally altered in materials at the border between solids and molecules. In magic sized clusters^{43–46} (MSC) containing a defined number of 26 atoms, a pronounced exciton-induced weakening of the crystal bonds occurs, resulting in a giant shift of the phonon spectrum driven by the exciton. As a consequence, a drastic increase of $\frac{dE_g}{dT}$ (up to a factor of 2) with respect to bulk material or nanocrystals of typical size is found. Using a first principles approach, we computed the exciton-induced change of the phonon density of states from molecular dynamics simulations on the excited state potential energy surface. This enabled us to quantitatively describe our experimental data by deriving the bandgap shift with temperature as the vibrational entropy contribution to the free energy difference between the ground and optically excited states.”

1. Han, P. & Bester, G. Large nuclear zero-point motion effect in semiconductor nanoclusters. *Phys. Rev. B* **88**, 165311 (2013).
2. Franceschetti, A. First-principles calculations of the temperature dependence of the band gap of Si nanocrystals. *Phys. Rev. B - Condens. Matter Mater. Phys.* **76**, 16–18 (2007).
3. Kamisaka, H., Kilina, S. V., Yamashita, K. & Prezhdo, O. V. Ab Initio Study of Temperature and Pressure Dependence of Energy and Phonon-Induced Dephasing of Electronic Excitations in CdSe and PbSe Quantum Dots †. *J. Phys. Chem. C* **112**, 7800–7808 (2008).
4. Ibrahim, Z. A. *et al.* Temperature dependence of the optical response: Application to bulk GaAs using first-principles molecular dynamics simulations. *Phys. Rev. B* **77**, 125218 (2008).

5. Capaz, R. B., Spataru, C. D., Tangney, P., Cohen, M. L. & Louie, S. G. Temperature dependence of the band gap of semiconducting carbon nanotubes. *Phys. Rev. Lett.* **94**, 1–4 (2005).
6. Han, P. & Bester, G. Fundamental difference between measured and calculated exciton-phonon coupling in nanostructures. *Phys. Rev. B* **99**, 100302 (2019).
7. Brooks, H. Theory of the Electrical Properties of Germanium and Silicon. in *Adv. Electr.* vol. 7 85–182 (1955).
8. Yang, J. *et al.* Route to the Smallest Doped Semiconductor: Mn²⁺-Doped (CdSe)₁₃ Clusters. *J. Am. Chem. Soc.* **137**, 12776–12779 (2015).
9. Landes, C. F., Braun, M. & El-Sayed, M. A. On the Nanoparticle to Molecular Size Transition: Fluorescence Quenching Studies. *J. Phys. Chem. B* **105**, 10554–10558 (2001).
10. Bawendi, M. G., Carroll, P. J., Wilson, W. L. & Brus, L. E. Luminescence properties of CdSe quantum crystallites: Resonance between interior and surface localized states. *J. Chem. Phys.* **96**, 946–954 (1992).
11. Puzder, A., Williamson, A. J., Gygi, F. & Galli, G. Self-healing of CdSe nanocrystals: First-principles calculations. *Phys. Rev. Lett.* **92**, 1–4 (2004).
12. Vörös, M., Galli, G. & Zimanyi, G. T. Colloidal Nanoparticles for Intermediate Band Solar Cells. *ACS Nano* **9**, 6882–6890 (2015).
13. Kovalenko, M. V *et al.* Prospects of Nanoscience with Nanocrystals. *ACS Nano* **9**, 1012–1057 (2015).
14. Akkerman, Q. A., Rainò, G., Kovalenko, M. V. & Manna, L. Genesis, challenges and opportunities for colloidal lead halide perovskite nanocrystals. *Nat. Mater.* **17**, 394–405 (2018).
15. Medintz, I. L., Uyeda, H. T., Goldman, E. R. & Mattoussi, H. Quantum dot bioconjugates for imaging, labelling and sensing. *Nat. Mater.* **4**, 435–446 (2005).
16. Choi, M. K., Yang, J., Hyeon, T. & Kim, D.-H. Flexible quantum dot light-emitting diodes for next-generation displays. *npj Flex. Electron.* **2**, 10 (2018).
17. Shirasaki, Y., Supran, G. J., Bawendi, M. G. & Bulović, V. Emergence of colloidal quantum-dot light-emitting technologies. *Nat. Photonics* **7**, 13–23 (2013).
18. García de Arquer, F. P., Armin, A., Meredith, P. & Sargent, E. H. Solution-processed semiconductors for next-generation photodetectors. *Nat. Rev. Mater.* **2**, 16100 (2017).
19. Efros, A. *et al.* Band-edge exciton in quantum dots of semiconductors with a degenerate valence band: Dark and bright exciton states. *Phys. Rev. B* **54**, 4843–4856 (1996).
20. Norris, D., Efros, A., Rosen, M. & Bawendi, M. Size dependence of exciton fine structure in CdSe quantum dots. *Phys. Rev. B. Condens. Matter* **53**, 16347–16354 (1996).
21. Ortner, G. *et al.* Temperature dependence of the excitonic band gap in In_xGa_{1-x}As/GaAs self-assembled quantum dots. *Phys. Rev. B* **72**, 085328 (2005).
22. Sanguinetti, S. *et al.* Electron-phonon interaction in individual strain-free GaAs/Al_{0.3}Ga_{0.7}As quantum dots. *Phys. Rev. B* **73**, 1–7 (2006).
23. Liptay, T. J. & Ram, R. J. Temperature dependence of the exciton transition in semiconductor quantum dots. *Appl. Phys. Lett.* **89**, 223132 (2006).
24. Joshi, A., Narsingi, K. Y., Manasreh, M. O., Davis, E. A. & Weaver, B. D. Temperature

- dependence of the band gap of colloidal CdSe/ZnS core/shell nanocrystals embedded into an ultraviolet curable resin. *Appl. Phys. Lett.* **89**, 89–92 (2006).
25. Valerini, D. *et al.* Temperature dependence of the photoluminescence properties of colloidal CdSe/ZnS core/shell quantum dots embedded in a polystyrene matrix. *Phys. Rev. B* **71**, 1–6 (2005).
 26. Jing, P. *et al.* Temperature-Dependent Photoluminescence of CdSe-Core CdS/CdZnS/ZnS-Multishell Quantum Dots. *J. Phys. Chem. C* **113**, 13545–13550 (2009).
 27. Nomura, S. & Kobayashi, T. Exciton–LO-phonon couplings in spherical semiconductor microcrystallites. *Phys. Rev. B* **45**, 1305–1316 (1992).
 28. Korsunskaya, N. E., Dybiec, M., Zhukov, L., Ostapenko, S. & Zhukov, T. Reversible and non-reversible photo-enhanced luminescence in CdSe/ZnS quantum dots. *Semicond. Sci. Technol.* **20**, 876–881 (2005).
 29. Olkhovets, A., Hsu, R.-C., Lipovskii, A. & Wise, F. Size-Dependent Temperature Variation of the Energy Gap in Lead-Salt Quantum Dots. *Phys. Rev. Lett.* **81**, 3539–3542 (1998).
 30. Wise, F. W. Lead Salt Quantum Dots: the Limit of Strong Quantum Confinement. *Acc. Chem. Res.* **33**, 773–780 (2000).
 31. Vossmeier, T. *et al.* CdS Nanoclusters: Synthesis, Characterization, Size Dependent Oscillator Strength, Temperature Shift of the Excitonic Transition Energy, and Reversible Absorbance Shift. *J. Phys. Chem.* **98**, 7665–7673 (1994).
 32. O'Donnell, K. P. & Chen, X. Temperature dependence of semiconductor band gaps. *Appl. Phys. Lett.* **58**, 2924–2926 (1991).
 33. Pässler, R. Dispersion-related description of temperature dependencies of band gaps in semiconductors. *Phys. Rev. B* **66**, 085201 (2002).
 34. Varshni, Y. P. Temperature dependence of the energy gap in semiconductors. *Physica* vol. 34 149–154 (1967).
 35. Allen, P. B. & Heine, V. Theory of the temperature dependence of electronic band structures. *J. Phys. C Solid State Phys.* **9**, 2305–2312 (1976).
 36. Heine, V. & Van Vechten, J. A. Effect of electron-hole pairs on phonon frequencies in Si related to temperature dependence of band gaps. *Phys. Rev. B* **13**, 1622–1626 (1976).
 37. Allen, P. B. & Cardona, M. Temperature dependence of the direct gap of Si and Ge. *Phys. Rev. B* **27**, 4760–4769 (1983).
 38. Ridley, B. K. *Quantum Process in Semiconductors*. (Oxford University Press).
 39. Giustino, F., Louie, S. G. & Cohen, M. L. Electron-phonon renormalization of the direct band gap of diamond. *Phys. Rev. Lett.* **105**, 1–4 (2010).
 40. Antonius, G., Poncé, S., Boulanger, P., Côté, M. & Gonze, X. Many-body effects on the zero-point renormalization of the band structure. *Phys. Rev. Lett.* **112**, 1–5 (2014).
 41. Saidi, W. A., Poncé, S. & Monserrat, B. Temperature Dependence of the Energy Levels of Methylammonium Lead Iodide Perovskite from First Principles. *J. Phys. Chem. Lett.* **7**, 5247–5252 (2016).
 42. Poncé, S. *et al.* Temperature dependence of the electronic structure of semiconductors and insulators. *J. Chem. Phys.* **143**, 102813 (2015).

43. Kasuya, A. *et al.* Ultra-stable nanoparticles of CdSe revealed from mass spectrometry. *Nat. Mater.* **3**, 99–102 (2004).
44. Yu, J. H. *et al.* Giant Zeeman splitting in nucleation-controlled doped CdSe:Mn²⁺ quantum nanoribbons. *Nat. Mater.* **9**, 47–53 (2010).
45. Cossairt, B. M. & Owen, J. S. CdSe Clusters: At the Interface of Small Molecules and Quantum Dots. *Chem. Mater.* **23**, 3114–3119 (2011).
46. Bowers, M. J., McBride, J. R. & Rosenthal, S. J. White-Light Emission from Magic-Sized Cadmium Selenide Nanocrystals. *J. Am. Chem. Soc.* **127**, 15378–15379 (2005).